# Effects of Deferasirox in Alzheimer’s Disease and Tauopathy Animal Models

**DOI:** 10.3390/biom12030365

**Published:** 2022-02-25

**Authors:** Ping Kwan, Amy Ho, Larry Baum

**Affiliations:** 1School of Pharmacy, The Chinese University of Hong Kong, Shatin, Hong Kong 999077, China; ping.kwan@sydney.edu.au (P.K.); ho3t@yahoo.com (A.H.); 2Level 5, R.M.C. Gunn Building, The University of Sydney, Sydney, NSW 2006, Australia; 3Department of Psychiatry, Li Ka Shing Faculty of Medicine, The University of Hong Kong, Hong Kong 999077, China

**Keywords:** tau, iron, chelator, transgenic, neurodegeneration, memory, plaque, tangle, desferrioxamine, Alzheimer’s disease

## Abstract

The accumulation of iron may contribute to Alzheimer’s disease (AD) and other tauopathies. The iron chelator desferrioxamine slows disease progression in AD patients. However, desferrioxamine requires injection, which is inconvenient and may hinder compliance. We therefore tested an oral iron chelator, desferasirox (Exjade), in transgenic animal models. Tg2576 mice overexpress the mutant human APP protein and produce the Aβ peptide. JNPL3 mice (Tau/Tau) overexpress the mutant human tau protein. Crossing these produced APP/Tau mice, overexpressing both APP and tau. Treating the three models with 1.6 mg deferasirox thrice weekly from age 8 to 14 months did not affect memory as measured by contextual fear conditioning or motor function as measured by rotarod, but tended to decrease hyperphosphorylated tau as measured by AT8 immunohistochemistry and immunoblotting. Deferasirox might act by decreasing iron, which aggregates tau, or directly binding tau to inhibit aggregation.

## 1. Introduction

Transition metals are implicated in the development of some neuropathological conditions, including Alzheimer’s disease (AD). Ions of Cu, Zn, Al, and Fe are capable of binding Aβ, inducing its protease resistance, reversible precipitation, O_2_-dependent production of H_2_O_2_, and concomitant toxicity [1,2,3,4,5,6].

The accumulation of iron may contribute to AD [5,7,8]. An iron-response element in the 5′ untranslated region (5′-UTR) of amyloid precursor protein (APP) mRNA induces translation, and iron chelators can decrease APP levels [9]. Higher than normal levels of iron were observed in AD brain samples, where iron colocalized with senile plaques, vessel walls, microglia, and neurofibrillary tangles (NFTs), which contain hyperphosphorylated tau protein [10,11,12,13]. Tau deficiency can lead to the accumulation of iron in cells [11]. Cognitive deterioration in AD correlates with both iron accumulation and NFTs, and iron is found in neurons that contain NFTs [14]. Fe^3+^ binds tau and induces its aggregation in NFTs and induction of heme oxygenase-1, which is upregulated in AD and may lead to the metabolism of heme from damaged mitochondria, releasing Fe^2+^ [11]. Fe^2+^ increases free radicals and can induce the aggregation of Aβ and hyperphosphorylated tau protein, while at the same time promoting the production of neurotoxic H_2_O_2_ with these protein aggregates in a mutually reinforcing (i.e., positive feedback) manner. Furthermore, reducing agents were shown to solubilize tau from its insoluble aggregated form, which may be due to the reduction of Fe^3+^ to Fe^2+^ [2,6,10,15,16,17]. By way of the Fenton reaction, amyloid binds Fe^3+^ and reduces it to Fe^2+^, and then binds O_2_ and converts it, using Fe^2+^, to H_2_O_2_ (Figure 1) [18]. Both the small protein aggregates (oligomers) of Aβ (as well as tau or other amyloid proteins) and reactive oxygen species (particularly H_2_O_2_) damage neurons and thus cognition [19,20]. Thus, iron is an interesting treatment target.

By decreasing the level of iron in animal models, iron-induced tau phosphorylation was reduced [21]. Iron chelation therapy for AD patients using desferrioxamine administered intramuscularly for 2 years significantly slowed disease progression [22]. Although results were promising, administration by injection may be uncomfortable and time consuming, thus reducing compliance. The more recently developed iron chelators, deferasirox (Exjade) and deferiprone (Ferriprox), may be more desirable since these drugs can be administered orally. An opinion paper suggested deferasirox for replacing desferrioxamine to treat AD [8]. Thus, we tested deferasirox for treating AD or tauopathies using animal models.

## 2. Materials and Methods

### 2.1. Animal Models and Treatments

Transgenic mouse models were bought (Taconic Farms) and bred at The Chinese University of Hong Kong (Shatin, Hong Kong, China) in standard housing conditions, and the University’s Animal Experimentation Ethics Committee approved the study. APP transgenic mice (Tg2576), which are hemizygous for the human β-amyloid precursor protein gene with the Swedish mutation (K670N/M671L) (APPswe), overexpress Aβ. They were bred on the F1 progeny of SJL/JcrNTac and C57BL6/NTac strains. Tau/Tau transgenic mice (JNPL3) were used as a tauopathy model. They are homozygous for the 4R0N isoform of the human microtubule-associated protein tau with the P301L mutation. Crossing JNPL3 with Tg2576 produced Tau/APP mice with one copy of the tau transgene and one copy of the APP transgene, as described previously [23,24]. We estimated 75% power for 10 mice of each strain, sex, and treatment using another Tg2576 chelator study, with an effect size of 1.24 for decreasing insoluble Aβ [1]. We did not pre-register the study.

Mice (Table 1) were randomly assigned to a treatment with or without 1.6 mg deferasirox, delivered in peanut butter thrice weekly from age 8 to 14 months. The dose was determined so that it was equivalent to a daily dose of 17–27 mg/kg (for mice weighing 40–25 g): within the 5–30 mg/kg daily range used in a study in humans [25], and tolerable when compared to 71 mg/kg daily used in a shorter-term (2 month) study in mice [26]. Behavioral and pathological assessors (below) were blind to the treatment of the mice.

### 2.2. Behavioral Tests

The mice were assessed by rotarod and contextual fear conditioning for motor and memory function, respectively. Rotarod [24] and contextual fear conditioning tests [27] were performed as previously described, both at the beginning and the end of the treatment period.

For the rotarod test, four rounds of trial training and one round of testing were performed with a one-hour interval in between. Mice were placed onto the rod of the apparatus. Rotation started at 4 revolutions per minute (rpm) and then increased at 1 rpm/9 s, to a maximum of 40 rpm. The time between the beginning of the acceleration and the falling of each mouse was recorded, thrice before treatment and thrice after treatment. The difference in the averages before vs. after treatment was calculated for each mouse.

Contextual fear conditioning (CFC) involves placing an animal in a new environment, giving an aversive stimulus, observing the response, waiting some time, and then putting the animal in the same environment but without an aversive stimulus to observe whether the animal exhibits fear (freezing its motion), indicating memory of the association of that environment with the aversive stimulus. Each mouse was placed in a chamber, and a tone was played, followed by an electric shock. The next day, context memory was tested by returning each mouse to the chamber and recording the time spent frozen. Cue memory was tested by moving the mouse to a novel chamber and recording the time spent frozen after the playing of the same tone as the previous day.

### 2.3. Immunohistochemistry

The distribution and phosphorylation of tau in the brains of transgenic mice (APP, Tau/Tau, and Tau/APP) was analyzed by immunohistochemistry. Briefly, coronal paraffin sections were dewaxed, rehydrated, boiled in citrate buffer (antigen retrieval), rinsed in 3% hydrogen peroxide (endogenous peroxidase quenching), rinsed with phosphate-buffered saline (PBS), blocked with 10% normal goat serum, and incubated overnight with mouse anti-tau-phospho-Ser202/Thr205 (1:50; AT8, MN1020, Thermo Scientific, Hong Kong, China) at room temperature. Sections were rinsed with PBS, incubated with DAKO Envision Labelled Polymer (Cat# K4001), rinsed, developed by DAB (Cat. 00-2114, Invitrogen, Hong Kong, China), rinsed with water, counterstained with Harris Hematoxylin, mounted, and examined by light microscope with a digital camera (Zeiss).

Two AT8 stained sections, about 80 μm apart, were quantitated. For each section, four fields were randomly selected from the medulla (about −5.68 to −6.00 mm from bregma) at 40× magnification based on the highest intensity of AT8 signals. Pixels within the color threshold were counted using SigmaScan Pro 5.0.

### 2.4. Western Blot

From each mouse brain, tissue was extracted as previously described [24]. Brainstem and spinal cord were used because they have relatively high concentrations of hyperphosphorylated tau. Tissue was homogenized in 3 volumes of cold homogenization buffer (150 mM Tris-HCl at pH 7.0, 2.25 M NaCl, 3 mM EGTA, 1.5 mM MgSO_4_, 6 mM dithiothreitol, 0.02% sodium azide, Protease Inhibitor Cocktail Set III (Cat. No. 539134, Calbiochem) at 1:100 dilution, and phosphatase inhibitor cocktail), centrifuged 15 min at 13,000× *g* and 4 °C, and supernatants removed and assayed to normalize sample concentrations. Normalized amounts were centrifuged for 60 min at 100,000× *g* and 4 °C. Pellets were homogenized in PHF extraction buffer (10 mM Tris-HCl at pH 7.4, 0.85 M NaCl, 10% sucrose, and 1 mM EGTA) and centrifuged for 30 min at 100,000× *g* and 4 °C. Supernatants were transferred to new tubes and incubated with sarkosyl (1%) for 1 h at 37 °C. After centrifugation at 100,000× *g* for 60 min at 4 °C, pellets were resuspended in Tricine sample buffer (100 mM Tris-HCl at pH 7.0, 4% SDS, 10% glycerol, 0.02% Coomassie Brilliant Blue G-500, 0.05% bromophenol blue, 0.005% phenol-chloroform, 2% beta-mercaptoethanol). Samples were analyzed by 10% sodium dodecyl sulfate polyacrylamide gel electrophoresis (SDS-PAGE) and AT8 Western blotting using enhanced chemiluminescence.

### 2.5. Statistical Analysis

Results were analyzed by SPSS Statistics (v20). Differences were considered significant if *p* < 0.05. Data were presented as mean ± standard deviation, and differences among groups were analyzed by independent sample *t* tests. The effect of the drug on mortality was analyzed by Fisher’s Exact Test.

## 3. Results

### 3.1. Mortality

A total of 106 mice entered treatment (Table 1). Only a small number of APP mice were available for study because breeding of this line was slow. Within the 6-month treatment period, 20 mice died: 9 with deferasirox treatment and 11 without. Stratifying by the three transgenic mouse models, deferasirox was not associated with survival.

### 3.2. Contextual Fear Conditioning

Contextual fear conditioning (CFC) tests measure memory of a location (context) or a sound (cue), expressed as the percentage of time spent frozen. By age 14 months, female Tau/Tau and Tau/APP mice had deteriorated to such a degree that they were unfit for CFC, thus only males were studied (Figure 2). For APP mice, both sexes were tested.

Between the start and end of treatment in Tau/Tau mice, CFC context memory changed −9.4 ± 27% without deferasirox and 2.9 ± 25% with deferasirox, *p* = 0.33, and CFC cue memory changed −16 ± 17% without deferasirox and −6.2 ± 37% with deferasirox, *p* = 0.47. Between the start and end of treatment in Tau/APP mice, CFC context memory changed −18 ± 29% without deferasirox and −8.9 ± 35% with deferasirox, *p* = 0.56, and CFC cue memory changed −33 ± 28% without deferasirox and −8.9 ± 20% with deferasirox, *p* = 0.062. Between the start and end of treatment in APP mice, CFC context memory changed −11 ± 20% without deferasirox and −19 ± 22% with deferasirox, *p* = 0.43, and CFC cue memory changed +2.8 ± 24% without deferasirox and −12 ± 25% with deferasirox, *p* = 0.19.

### 3.3. Rotarod

Rotarod tests measure motor ability and coordination. At age 14 months, female Tau/APP mice had deteriorated to such a degree that they were not fit for rotarod tests, thus only males were studied (Figure 3). For Tau/Tau and APP mice, both sexes were tested.

Between the start and end of treatment in Tau/Tau mice, the rotarod times changed −16 ± 65 s without deferasirox and −13 ± 30 s with deferasirox, *p* = 0.94. Between the start and end of treatment in Tau/APP mice, the rotarod times changed +11 ± 43 s without deferasirox and +11 ± 45 s with deferasirox, *p* = 0.96. Between the start and end of treatment in APP mice, the rotarod times changed +3.5 ± 29 s without deferasirox and +3.5 ± 37 s with deferasirox, *p* = 1.00.

### 3.4. AT8 Immunohistochemistry

The area occupied by NFTs in mouse brains (males and females combined) was analyzed by immunohistochemistry using monoclonal antibody AT8, which targets Ser202 and Thr205 of the hyperphosphorylated tau in NFTs (Figure 4). In Tau/Tau mice, NFTs occupied 5.4 ± 2.7% of area without deferasirox treatment and 4.0 ± 2.6% with deferasirox: a 25% decrease, *p* = 0.22. In Tau/APP mice, NFTs occupied 4.5 ± 2.6% without deferasirox treatment and 3.8 ± 3.4% with deferasirox: a 15% decrease, *p* = 0.57.

Only a small number of APP mice were available for study. Thus, analysis of their brains was not undertaken.

### 3.5. AT8 Western Blot

From each brain, a fraction enriched in paired helical filaments (PHF) was extracted and subjected to AT8 Western blotting to measure accumulation of the 64 kD band of phosphorylated tau (Figure 4 and Appendix A). Only males were studied because only a few female brains were available.

In Tau/Tau mice, the level of phosphorylated tau was 24 ± 17 (arbitrary units) without deferasirox and 18 ± 17 with deferasirox: a 24% decrease, *p* = 0.64. In Tau/APP mice, the level of phosphorylated tau was 26 ± 12 without deferasirox and 12 ± 14 with deferasirox: a 53% decrease, *p* = 0.03.

## 4. Discussion

In the particular conditions of this study, deferasirox treatment of AD and tauopathy models did not demonstrate protection of memory or motor ability. However, there is a possibility that deferasirox decreased hyperphosphorylated tau accumulation, as indicated by decreased hyperphosphorylated tau on Western blots of Tau/APP mice. This is consistent with trends toward decreased hyperphosphorylated tau on Western blots of Tau/Tau mice and in immunohistochemistry of both Tau/Tau and Tau/APP mice.

Consistent with a possible protective effect of deferasirox in tauopathy are the trends toward relatively better memory, in both CFC cue and context tests, in both Tau/Tau and Tau/APP mice. However, because the p-values were high and because many comparisons were tested in this study, replication of these findings would be needed.

Although not statistically significant, it may be worth noting the trend toward worsening of memory with deferasirox in APP mice. Perhaps deferasirox acts directly or indirectly on tau, but not on Aβ, and removal of iron to levels below the optimum may impair function in the absence of a specific benefit. However, human studies of deferasirox have not led to cognitive decline [28].

If deferasirox does reduce hyperphosphorylated tau levels, there are several possible mechanisms. Iron induces aggregation of hyperphosphorylated tau, thus such an effect might be due to iron chelation [15]. Another possibility is that deferasirox directly binds tau, blocking its aggregation. There are many molecules that interfere with tau aggregation, and some of these compounds share structural similarities with deferasirox [29,30,31]. There is precedent for other chelators, such as curcumin and clioquinol, to prevent neuropathology or behavioral deterioration in transgenic APP mice [1,32]. As with deferasirox, it is possible that these drugs slow neurodegeneration by pathways other than chelation, such as by curcumin directly inhibiting Aβ aggregation [33].

These results suggest fruitful directions for future research. Studies of deferasirox in larger numbers of animals should be conducted to determine whether this drug has the potential to treat tauopathies. Epidemiological studies may investigate whether long-term users of deferasirox and other iron chelators display a lower risk of developing tauopathies (though interpretation may be complicated if a high level of iron, which is of course common among chelator users, itself raises the risk of tauopathies). Clinical trials in people genetically prone to later develop tauopathy could explore whether iron chelators slow or prevent tau deposition and neurodegeneration.

## Figures and Tables

**Figure 1 biomolecules-12-00365-f001:**
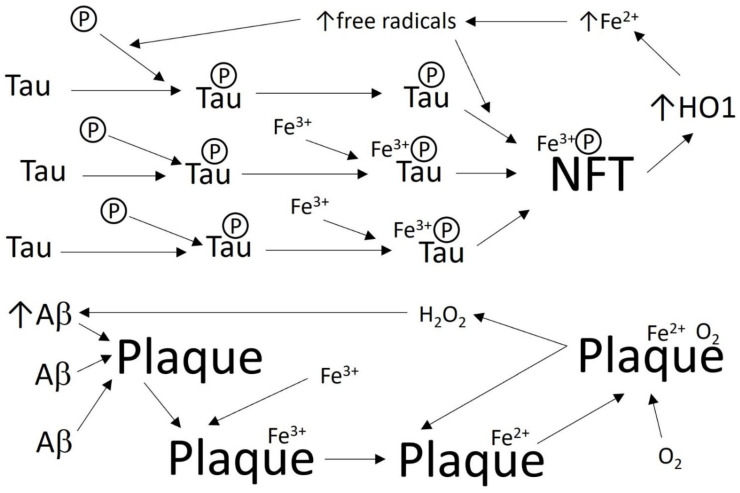
Some interactions of iron with AD pathology. Iron may increase in AD, accumulating on plaques and NFTs [16,18]. In AD, some tau is hyperphosphorylated (a phosphate group is shown as a circled P). Fe^3+^ can bind tau and increase its aggregation and induction of heme oxygenase 1 (HO1), which raises Fe^2+^, which in turn raises free radicals, which then increases tau aggregation, for example by stimulating tau phosphorylation [16,17]. Aβ aggregates to form plaques, and iron on plaques makes H_2_O_2_ by the Fenton reaction, as follows: Fe^3+^ binds plaque Aβ, which reduces Fe^3+^ to Fe^2+^, and then binds O_2_ and converts it, using the Fe^2+^, to H_2_O_2_ [18]. In a positive feedback loop, H_2_O_2_ increases Aβ [10].

**Figure 2 biomolecules-12-00365-f002:**
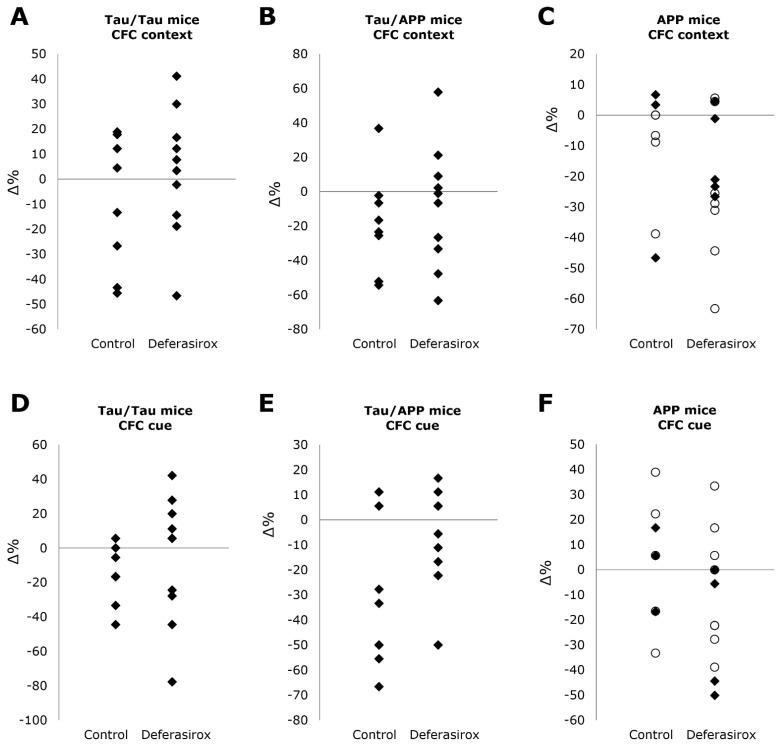
Deferasirox did not significantly affect differences in percentage of time motionless (vertical axis) in contextual fear conditioning (CFC) tests between the start and end of treatment for context (location) in (**A**) Tau/Tau (*p* = 0.33), (**B**) Tau/APP (*p* = 0.56), or (**C**) APP (*p* = 0.43) mice, or for cue (sound) in (**D**) Tau/Tau (*p* = 0.47), (**E**) Tau/APP (*p* = 0.062), or (**F**) APP (*p* = 0.19) mice. Males are black diamonds, and females are white circles. By age 14 months, female Tau/Tau and Tau/APP mice had deteriorated to such a degree that they were unfit for CFC tests, thus only males were studied.

**Figure 3 biomolecules-12-00365-f003:**
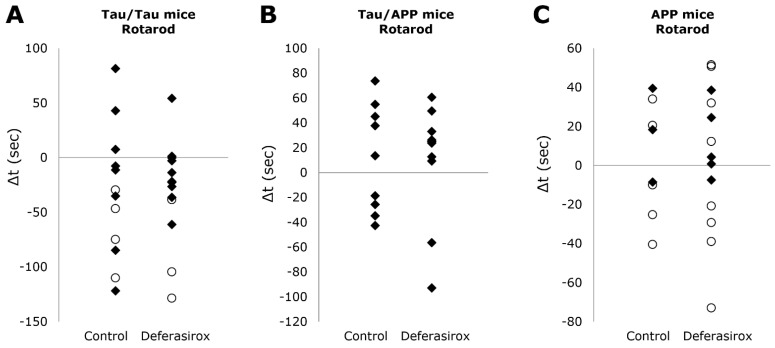
Deferasirox did not significantly affect differences in the time mice stayed on the rotarod (seconds, vertical axis) between the start and end of treatment for (**A**) Tau/Tau (*p* = 0.94), (**B**) Tau/APP (*p* = 0.96), or (**C**) APP (*p* = 1.00) mice. Males are black diamonds, and females are white circles. By age 14 months, female Tau/APP mice had deteriorated to such a degree that they were not fit for rotarod tests, thus only males were studied.

**Figure 4 biomolecules-12-00365-f004:**
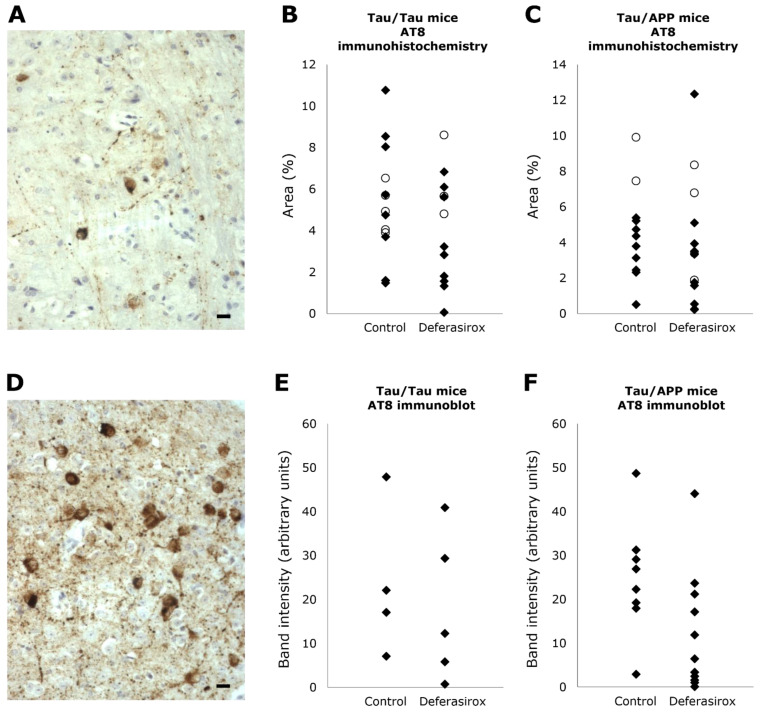
AT8 antibody immunoreactivity. Examples (**A**,**D**) of AT8 immunoreactivity in coronal medulla sections of Tau/APP mice treated with (**A**) or without deferasirox (**D**) (scale bar is 25 µm). Comparing immunoreactivity of deferasirox-treated mice with untreated mice, the mean area occupied by neurofibrillary tangles in medulla sections was 25% less in Tau/Tau mice (**B**, *p* = 0.22) and 21% less in Tau/APP mice (**C**, *p* = 0.57), and the immunoblot band intensity was 24% less in Tau/Tau mice (**E**, *p* = 0.64) and 53% less in Tau/APP mice (**F**, *p* = 0.034). Males are black diamonds, and females are white circles. Because few female brains were available for homogenization, only male samples were immunoblotted.

**Table 1 biomolecules-12-00365-t001:** Numbers of mice entering and completing the study. Denominators are the numbers of mice entering the study. Numerators are the numbers of mice completing the study.

	Tau/Tau		Tau/APP		APP	
	Deferasirox	Control	Deferasirox	Control	Deferasirox	Control
Male	10/10	8/10	11/12	9/13	5/5	3/3
Female	6/10	9/9	6/10	6/9	8/8	5/7
Total	16/20	17/19	17/22	15/22	13/13	8/10

## Data Availability

The data that support the findings of this study are available from the corresponding author upon reasonable request.

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
