# Peer review of "Effects of Deferasirox in Alzheimer’s Disease and Tauopathy Animal Models"

_biomolecules, 2022, doi:10.3390/biom12030365_

Round 1
Reviewer 1 Report
Paper is well-written and will be interesting to the readers of Biomolecules. Paper could be accepted with moderate grammar corrections.
Author Response
Response: Thank you for your positive comments. We have made the following grammar corrections:
- Line 30: Spelled out 5'-UTR as 5’ untranslated region.
- Line 36-43: Split this long sentence into 2.
- Line 71: Added “, which are”.
- Line 73: Added a comma.
- Line 122: Corrected the distance to “mm”.
- Lines 251-2: Changed to “iron chelators slow or prevent”.
Reviewer 2 Report
This article entitled “Effects of deferasirox in Alzheimer’s disease and tauopathy animal models”, describes effect of deferasirox on iron levels, which aggregates tau, or tau aggregation inhabitation. Overall, this reviewer feels this is an interesting study and suggests acceptance of this article to vaccines with minor revision.
The crucial point is rewriting the introductory part by proper connection and relevant references for Iron, hydrogen peroxide and their role in AD via Fenton reaction. This will give readers a proper idea of Iron levels and their role in AD. Even one general figure with all these factors would be of great value.
Author Response
Response: Thank you for your positive comments. We have added a figure and information to the second paragraph of the introduction on roles of iron and hydrogen peroxide in AD via Fenton reaction, as follows:
- Lines 36-9: Changed to “Fe3+ binds tau and induces its aggregation in NFTs and induction of heme oxygenase-1, which is upregulated in AD and may lead to metabolism of heme from damaged mitochondria, releasing Fe2+ [11].”
- Lines 39-40: Changed to “Fe2+ increases free radicals and can induce aggregation of Aβ and hyperphosphorylated tau protein”.
- Lines 43-5: Added “By Fenton reaction, amyloid binds Fe3+ and reduces it to Fe2+, and then binds O2 and converts it, using Fe2+, to H2O2 (Figure 1) [18].”
- Figure 1: We added a figure to show some interactions of iron with AD pathology, including tau, NFT, plaques, hydrogen peroxide, and Fenton reaction.
Reviewer 3 Report
The manuscript "Effects of deferasirox in Alzheimer’s disease and tauopathy animal models" by Kwan et al. reports the research on the influence of deferasirox (an oral iron chelator, already used in the treatment of iron overload) on behavior and memory, as well as on hyperphosphorylated tau protein in brains of transgenic mice being the model for Alzheimer's disease. The results suggest that deferasirox may decrease the levels of hyperphosphorylated tau in mice brains. Although the reader cannot say that other parameters were improved due to treatment, it is essential to publish such results in case other researchers want to optimize this research.
Minor suggestions:
- It could be beneficial to read in the manuscript the reasoning for the chosen dose of the drug.
- Abstract, line 14: it would be useful to add the word "human" before APP and tau. It is crucial because mouse and human APP and tau have different sequences and, thus, properties.
- Line 107: the unit of the distance between sections should be corrected.
Author Response
Response: Thank you for your positive comments. We have made the following changes:
- Section 2.1, paragraph 2, lines 82-5: We added a rationale for choosing the dose: “The dose was determined so that it was equivalent to a daily dose of 17-27 mg/kg (for mice weighing 40-25 g): within the 5-30 mg/kg daily range used in a study in humans [25], and tolerable when compared to 71 mg/kg daily used in a shorter-term (2 month) study in mice [26].”
- Lines 14-5: We added the word “human” before APP and tau.
- Line 122: We corrected the unit of distance between sections.